# Electrospun Water-Borne Polyurethane Nanofibrous Membrane as a Barrier for Preventing Postoperative Peritendinous Adhesion

**DOI:** 10.3390/ijms20071625

**Published:** 2019-04-01

**Authors:** Shih-Heng Chen, Pang-Yun Chou, Zhi-Yu Chen, Feng-Huei Lin

**Affiliations:** 1Institute of Biomedical Engineering, College of Medicine and College of Engineering, National Taiwan University, Taipei 100, Taiwan; shihheng@mac.com (S.-H.C.); d79010340217@gmail.com (Z.-Y.C.); 2Department of Plastic and Reconstructive Surgery, Chang Gung Memorial Hospital, Chang Gung University and Medical College, Taoyuan 333, Taiwan; chou.asapulu@gmail.com; 3Division of Biomedical Engineering and Nanomedicine Research, National Health Research Institutes, Miaoli 35053, Taiwan

**Keywords:** peritendinous adhesion, tendon adhesion, antiadhesion, adhesion barrier, tendon repair, water-borne polyurethane, electrospun

## Abstract

Peritendinous adhesion is a major complication after tendon injury and the subsequent repairs or reconstructions. The degree of adhesion can be reduced by the interposition of a membranous barrier between the traumatized tendon and the surrounding tissue. In the present study, electrospun water-borne polyurethane (WPU) nanofibrous membranes (NFMs) were created for use after the reparation or reconstruction of tendons to reduce adhesion. In the electrospinning process, water was employed as the solvent for WPU, and this solvent was ecofriendly and nontoxic. The nanofibrous architecture and pore size of the WPU NFMs were analyzed. Their microporosity (0.78–1.05 µm) blocked the penetration of fibroblasts, which could result in adhesion and scarring around the tendon during healing. The release of WPU mimicked the lubrication effect of the synovial fluid produced by the synovium around the tendon. In vitro cell studies revealed that the WPU NFMs effectively reduced the number of fibroblasts that became attached and that there was no significant cytotoxicity. In vivo studies with the rabbit flexor tendon repair model revealed that WPU NFMs reduced the degree of peritendinous adhesion, as determined using a gross examination; a histological cross section evaluation; and measurements of the range of motion of interphalangeal joints (97.1 ± 14.7 and 79.0 ± 12.4 degrees in proximal and distal interphalangeal joints respectively), of the length of tendon excursion (11.6 ± 1.9 cm), and of the biomechanical properties.

## 1. Introduction

Tendons are formed from collagenous tissue that connects muscles to bones in limbs. Tendons cross the joints in the extremities to facilitate flexion and extension during muscle contraction. Tendon injuries are common in trauma to extremities and require a delicate surgical exploration and the meticulous repair of the ruptured tendons [1,2]. To achieve optimal outcomes, early tendon repair followed by immobilization with protective splinting for 4 weeks and a regular rehabilitation program for 6 to 8 weeks [3] are mandatory. However, the risk of tendon adhesion is higher in severely traumatized wounds [4]. Adhesion and scar formation are commonly related to tissue injury, foreign body reaction, infection, bleeding, and ischemia [5] and are a major component of the normal healing process. Thus, during wound and tendon healing, fibrin deposition and scar formation are inevitable and contribute to the subsequent tendon adhesion. Minor adhesion can be overcome through rehabilitation. However, severe adhesion and scars can cause serious complications, including a limited range of motion, pain, and even the requirement of secondary operations such as tenolysis and scar release [4,6,7]. Among all tendon injuries, traumas involving the flexor digitorum profundus (FDP) or flexor digitorum superficialis tendons in zone II of the hand are those most frequently complicated by peritendinous adhesion [8]. The pathophysiology of adhesion formation following tendon injury and its subsequent repair is still not fully understood but is believed to be related to the extrinsic healing process led by local fibroblasts [9].

An ideal barrier used to prevent or reduce the degree of adhesion around tendons should mimic the physiological features of the tendon sheath, which is a membranous layer outside of the tendon that separates the tendon from surrounding tissue. The functions of the tendon sheath include the maintenance of an efficient route for tendon excursion and the secretion of synovial fluid for lubrication during gliding [10,11]. The synovial fluid secreted by the tendon sheath not only lubricates the tendon but also acts as an interface across which nutrients and waste diffuse [10,11]. Thus, a membrane employed to suppress peritendinous adhesion should be physiologically compatible and have the following features: (1) serve as a barrier to prevent fibroblast penetration; (2) not impede the diffusion of nutrients and waste; and (3) facilitate tendon gliding. These objectives can be achieved by using a nanofibrous membrane (NMF). An NFM effectively separates tissues through its submicrometer pore size and prevents fibroblast penetration and subsequent adhesion. However, because of its high porosity, the membrane does not compromise the exchange of nutrients and waste.

Electrospinning is a flexible method that has been employed to produce NFMs based on either natural or artificial polymers [12]. NFMs composed of poly-caprolactone, poly(lactide-co-glycolide), polylactide-poly(ethylene glycol) tri-block copolymer, or a blend of chitosan and alginate have been demonstrated to prevent adhesion [13,14,15]. However, these NFMs were investigated with only in vivo abdominal antiadhesion models and not applied to tendons. NFMs designed to prevent peritendinous adhesion have recently been reported, including those based on hyaluronic-acid-loaded polycaprolactone [16,17] and a poly(l-lactic acid)-polyethylene glycol polymer loaded with ibuprofen [18]. In the present study, our purpose was to create an electrospun NFM that exerts a lubricating effect and has elastomeric properties to facilitate an intraoperative application, a stable structure for 4–6 weeks before degradation, biocompatibility and biodegradability, and no toxicity during its production. Water-borne polyurethane (WPU) met our requirements. During the electrospinning of WPU, water rather than organic solvent is used; this eliminates the concerns of toxicity and pollution associated with the use of conventional solvent-based polyurethane.

In this study, an electrospun NFM based on WPU (a WPU NFM) was produced. In vitro fibroblast studies were employed to evaluate the attachment to the NFM as well as cytotoxicity. In vivo studies using a rabbit hind-paw tendon repair model were used to reveal the efficacy of the WPU NFM in preventing peritendinous adhesion. The parameters evaluated were gross inspection, histological analysis, range of motion of joints, length of tendon excursion, and biomechanics. The effect of the WPU NFM was compared with not only that of a surgical control treatment but also that of the use of Seprafilm^®^, a commercial product for the prevention of adhesion after abdominal surgery.

## 2. Results

### 2.1. Preparation and Characterization of the Electrospun NFM

In the electrospinning process of WPU dispersion, poly-ethylene-oxide (PEO), a water-soluble polymer, was added. In one study [19], a minimum PEO concentration of 4 wt% in the water was required to successfully obtain electrospun WPU fibers. Thus, in our study, a slightly higher PEO concentration of 5.66 wt% was selected for the dispersion preparation. Different WPU/PEO compositions were blended to determine the optimal proportion. Constant and stable fibers were obtained from all blends except that with a 3/2 (*v*/*v*) composition (hereafter named W3P2). For this W3P2 composition, the electrospinning process was unstable, probably because the viscosity of the blend was relatively high. In addition, the electrospun fibers obtained using this blend contained beads (as shown in Figure 1), and the material was relatively fragile. Figure 1 (upper row) shows scanning electron microscopy (SEM) images of the electrospun fibers obtained using different compositions. SEM images of the fibers after the removal of PEO through a treatment with phosphate-buffered saline (PBS) are also presented in Figure 1 (lower row). When the 1/1 WPU/PEO composition (W1P1) was employed, no beads formed and the shape of the fibers was unaffected by the PBS treatment, indicating the stability of the fibers. 

The diameters of fibers obtained using different WPU/PEO compositions ranged from 349.5 to 369.3 nm, with the diameter being largest for the W1P1 composition and lowest for W2P3 (Table 1). This further supports the high stability of the W1P1 fibers. In addition, using SEM measurements, the average pore size was discovered to be negatively correlated with the PEO proportion. The higher the PEO proportion, the smaller the pores were (Table 1). The pore sizes ranged from 0.78 to 1.05 μm, depending on the composition (Figure 1). Because fibroblasts generally have diameters larger than 8 μm, all of the NFMs could theoretically block fibroblast penetration with their micrometer pores (0.78–1.05 μm). These pores also allowed the diffusion of nutrients required for tendon healing. As for the pore sizes among different groups, W3P2 exhibited significantly larger pores (1.05 ± 0.44) when compared to W1P2 (0.78 ± 0.31 m) and W2P3 (0.84 ± 0.31 m), with *p*-values of 0.002 and 0.012, respectively. The pore sizes of W1P1 (0.95 ± 0.36 m) did not show a significant difference when compared to the other three groups.

To remove PEO in obtaining pure WPU fibers, the electrospun fibers were washed with a PBS solution. The extraction of PEO was monitored using Fourier transform infrared spectroscopy (FTIR). Figure 2 presents the FTIR spectra of one of the investigated compositions (W1P1) before and after PBS treatment. In the spectrum obtained before the PBS wash (depicted as W1P1BW in Figure 2), there was a band centered at approximately 1120 cm^−1^, which was attributed to the C–O–C stretching vibration of PEO. This band was absent in the spectrum obtained after the fibers were treated with PBS (depicted as W1P1 in Figure 2). In addition, the spectrum of the fibers after PBS treatment (W1P1) revealed the same absorption as that in the spectrum of pure WPU (depicted as WPU in Figure 2). On the basis of the FTIR findings, we infer that PEO could be completely removed from the electrospun fibers through PBS treatment. 

### 2.2. Mechanical Properties

Table 2 lists the mechanical properties of the fibers, namely the ultimate stress, ultimate strain, and Young’s modulus calculated from the stress–strain curves. The Young’s modulus, or elastic modulus, defines the ratio of tensile stress to tensile strain along an axis and indicates the stiffness of an elastic material. Mechanical properties are crucial in an NFM designed to prevent peritendinous adhesion because the NFM must stretch and bear tension when it is wrapped around a repaired tendon during surgery. The NFMs with different volume WPU/PEO ratios have similar stress–strain curves with an initial elastic region and ultimate failure (Figure 3). After PEO was removed from the NFMs through PBS treatment, the ultimate tensile strength, the elongation percentage at break, and Young’s modulus of the NFMs became significantly higher (Figure 3 and Table 2), with a *p*-value < 0.05. Although W1P2 NFM revealed the highest Young’s modulus (0.92 MPa), its ultimate tensile strength (4.80 MPa) and elongation percentage at break (236.5%) were relatively poor compared with the rest (with *p*-value < 0.05); therefore, W1P2 was not considered an optimal composition. W3P2 had the highest WPU composition, but it was ruled out because of its instability during the electrospinning process. The W1P1 and W2P3 NFMs had similar elongation percentages at break (432.4% and 445.0% respectively, with *p*-value > 0.05); although W2P3 revealed a higher ultimate tensile strength (8.58 MPa versus 7.25 MPa, with *p*-value< 0.05), we decided to use W1P1 as the optimal WPU/PEO composition because of its higher proportion of WPU.

### 2.3. In Vitro Degradation

The in vitro degradation test was performed at a temperature of 37 °C. As illustrated in Figure 4, W3P2 had the highest proportion of WPU and, thus, degraded the slowest, whereas W1P2 degraded the quickest. W1P1 degradation required approximately 100 days under the in vitro conditions. This implies that the W1P1 NFM may survive for at least 100 days after its insertion following tendon surgery, which would exert a sufficient antiadhesive effect during tendon healing and immobilization after tendon repair.

### 2.4. Cytotoxicity and Cell Attachment Test

Regarding cytotoxicity, the 3-(4,5-dimethylthiazol-2-yl)-2,5-diphenyltetrazolium bromide (MTT) assay results were presented in Figure 5A. In the MTT assay, human dermal fibroblasts were use because of two reasons: 1. peritendinous adhesions are related to the extrinsic healing process mediated by extrinsic fibroblasts outside the traumatized tendon, and 2. using human dermal fibroblasts mimics clinical scenarios. The cell viability of W3P2, W1P1, W2P3, W1P2, and the negative control were 88.6%, 89.4%, 93.6%, 94.3%, and 87.4%, respectively, when defining the control group as 100%. The data reveals that the WPU NFMs with different compositions were not toxic to human dermal fibroblasts (HDFs).

To evaluate fibroblast attachment on NFMs with different WPU/PEO compositions, HDFs were seeded on these membranes (day 0) and inoculated for 1 and 4 days. The number of HDFs attached to the surface of WPU NFMs and tissue culture polystyrene (TCPS) as well as Seprafilm was evaluated using MTT assays (Figure 5B). On day 0, there were 8.5 × 10^3^ cells attached to TCPS and 3.6 × 10^3^ cells to Seprafilm, while there were 1.5 × 10^3^, 2.0 × 10^3^, 1.6 × 10^3^, and 1.9 × 10^3^ cells attached to W3P2, W1P1, W2P3, and W1P2 NFMs, respectively. On day 1, the cell number increased to 9.6 × 10^3^ on TCPS and to 3.9 × 10^3^ on Seprafilm compared to 1.8 × 10^3^, 2.0 × 10^3^, 2.0 × 10^3^, and 1.9 × 10^3^ cells on W3P2, W1P1, W2P3, and W1P2 NFMs respectively (*p* < 0.05 when compared to TCPS; *p* < 0.05 when compared to Seprafilm). On day 4, the cells on TCPS increased to 2.7 × 10^4^, while those on W3P1, W1P1, W2P3, and W1P2 NFMs increased to 3.1 × 10^3^, 4.0 × 10^3^, 3.5 × 10^3^, and 3.8 × 10^3^ respectively (*p* < 0.05 compared to TCPS). Figure 5B reveals that significantly fewer HDFs were attached to the WPU NFMs than the TCPS on days 0, 1, and 4, regardless of the WPU/PEO proportions. When compared to Seprafilm, all WPU NFMs exhibited fewer cell attachment on day 1. These findings indicate that the WPU NFMs effectively reduced the amount of fibroblast attachment. When comparing NFMs with different WPU/PEO compositions, there was no significant difference between each group (*p* > 0.05). In addition, the number of HDFs on the WPU NFMs had about a 2.0–2.1-fold increase from day 0 to day 4, which further implies that the WPU NFMs were not cytotoxic and that fibroblasts could grow on them. The low cell number was related to the low number of initially attached cells.

SEM images of the HDFs attached to NFMs were obtained (Figure 5, lower rows). The HDFs occupied a small area on all WPU NFMs. This SEM finding echoes the quantitative data of the few attached cells presented in Figure 5B. The W1P1 NFM was selected for further in vivo investigation because no beads formed in the NFM during electrospinning, and the NFM demonstrated only small variations in fiber diameter, favorable mechanical properties, and low-level fibroblast attachment.

### 2.5. Animal Study

#### 2.5.1. Gross Evaluation

The severity of adhesion around the repaired tendon was evaluated through a direct observation 3 weeks after the initial procedure. Figure 6 presents the gross photographs of the repaired tendons in different groups. Dense scarring and considerable adhesion were noted around the repaired tendon, especially at the suture site, in the surgical control group; a sharp dissection was required to divide the excessive fibrous tissue and, thus, to free the tendon. In the Seprafilm group, moderate adhesion and scarring were discovered between the repair site of the tendon and its surrounding tissue; however, the adhesion was not as dense as that in the control group and could be separated through a blunt dissection. In the group treated with the W1P1 NFM after tendon repair, the surface of the tendon was relatively smooth; moreover, the adhesion was mild compared with that in the surgical control and Seprafilm groups.

#### 2.5.2. Histology

Representative histological sections were stained with hematoxylin and eosin (H&E) and Masson trichrome blue. Tendons wrapped with WPU NFMs were compared with those in the surgical control and Seprafilm groups (Figure 7). In the surgical control group, considerable adhesion was noted between the repaired tendon and surrounding tissue, with the interface occupied by scar tissue. For tendons wrapped with Seprafilm, loose fibrous tissue was noted to bridge the repaired tendon with the surrounding tissue, resulting in the repaired tendon having a rough surface; this indicated a moderate infiltration of fibroblasts and the resultant adhesion formation. By contrast, less adhesion was observed in the tendons wrapped with the W1P1 NFM. In the W1P1 group, the surface of the repaired tendon was smooth, and a clear interface between the tendon and its surrounding tissue could be identified, indicating a favorable healing with little adhesion associated with extrinsic fibroblasts. The W1P1 NFM was, thus, discovered to be a superior adhesion barrier to Seprafilm in a gross evaluation (Figure 6) and a histological section examination (Figure 7).

#### 2.5.3. Range of Motion

To objectively quantify the antiadhesion efficacy of the NMF in vivo, the range of motion of the distal interphalangeal (DIP) and proximal interphalangeal (PIP) joints and an excursion of the repaired FDP tendons were evaluated and the biomechanical measurements were performed, including for the pullout force and breaking force in the surgical control, Seprafilm, and W1P1 NFM groups (Figure 8). These parameters were measured because of their clinical relevance to peritendinous adhesion [20,21].

The range of motion of the PIP and DIP joints in the W1P1 NFM group were 97.1 ± 14.7 and 79.0 ± 12.4 degrees, respectively, while that in the Seprafilm group were 55.6 ± 7.3 and 46.3 ± 9.9 degrees and that in the control group were 48.8 ± 14.6 and 35.6 ± 13.2 degrees (Figure 8A,B). The W1P1 group exhibited a significantly better range of motion compared to the other two groups (*p* < 0.05). 

The length of the FDP tendon excursion of the W1P1 NFM group was 11.6 ± 1.9 cm, which was significantly better than that of the Seprafilm (7.5 ± 1.7 cm) and surgical control (7.3 ± 1.7 cm), with *p*-value < 0.05 (Figure 8C). The quantitative evaluation of the antiadhesive properties revealed that the W1P1 NFM had the most favorable effect, followed by Seprafilm and then the control.

#### 2.5.4. Pullout and Breaking Force

The pullout force is defined as the minimal force required to completely pull the FDP tendon out of its tendon sheath and surrounding tissue; it indicates the severity of adhesion around a tendon. As displayed in Figure 8D, the greatest force required to pull the FDP tendon out of its surrounding tissue was noted in the surgical control group (5.5 ± 1.4 N), implying a high severity of adhesion. The least force was required with tendons wrapped with the W1P1 NFM (3.3 ± 1.5 N), whereas a moderate force was required in the Seprafilm group (5.0 ± 2.3 N). These findings suggest that the W1P1 NFM exerted the most favorable antiadhesive effect (*p* < 0.05).

The breaking force is defined as the minimal force required to tear a tendon by pulling both ends until a rupture occurs. The breaking force is positively correlated with the healing of a tendon. As illustrated in Figure 8E, the breaking force was slightly higher in the W1P1 group (6.8 ± 1.4 N) than in the other two groups (Seprafilm was 6.2 ± 3.3 N and control was 5.3 ± 2.2 N), but the difference was nonsignificant. Because the tendons wrapped with the W1P1 NFM had a breaking strength comparable to that of the untreated controls, we concluded that the W1P1 NFM did not have any significant adverse effect on the healing process of tendons. 

Overall, our results demonstrated that the W1P1 NFM is a favorable tendon barrier for preventing adhesion after tendon surgery and does not hinder tendon healing.

## 3. Discussion

Tendon barriers are employed to reduce scar or adhesion formation between an injured tendon and surrounding tissues [22]. Ideally, the barrier should be not only biocompatible but also able to maintain its form and structure in the body until the repaired tendon has healed (approximately 4 to 6 weeks) and until the rehabilitation process has begun to reduce the degree of adhesion and scarring [23]. Additionally, the barrier must be flexible and slightly elastic to enable adequate wrapping around the cylindrically shaped tendons during surgical repair [24,25]. US Food and Drug Administration (FDA)-approved adhesion barriers such as Seprafilm™ and SurgiWrap™ are designed to prevent peritoneal adhesion after abdominal surgery rather than peritendinous adhesion after tendon surgery. At the end of laparotomy, Seprafilm™, which is a thick membrane of sodium hyaluronate and carboxymethylcellulose, can be applied to the surface of the peritoneum to keep it anatomically separated from the abdominal wall [26]. Seprafilm™ degrades within approximately 1 week in vivo. However, because the healing process of tendons is slow and immobilization is necessary for 4 to 6 weeks, Seprafilm™ is not an ideal adhesion barrier for preventing peritendinous adhesion. SurgiWrap™ is a dense membrane composed of polylactides [27]. Its hydrophobic character could reduce the degree of fibroblast adhesion, but its thickness may hinder the exchange of nutrients and waste in the trauma zone, potentially leading to interference with tendon healing. In brief, because both adhesion barriers are dense membranes, the diffusion and exchange of nutrients and waste could be compromised. This could negatively affect the healing of tendons because blood circulation in tendons is limited and tendon healing is greatly dependent on diffusion to exchange nutrients and waste. In addition, because both barriers are thick and relatively inflexible, they are more suitable for application to a flat surface, such as the peritoneum. For circumstances that require wrapping, such as around a cylindrically shaped tendon, these products are more difficult to use. 

Polyurethane (PU) is a segmented polymer with a microphase-segmented morphology, which is reflected by its elastomeric behavior. The properties of PU can be tailored by a modification of its composition for different purposes, such as for materials used in dressing wounds [28,29,30,31,32]. Electrospun nanofibers composed of polycaprolactone-based PU are widely applied in tissue engineering [33,34] and drug delivery [35] because of their biodegradability, favorable mechanical properties, and FDA approval. Segmented amino acid–based PU with elastomericity and biodegradability was developed by modifying the structure to contain poly(l-tyrosine) [36]. Aligned and randomly braided biodegradable culture scaffolds based on PU were produced through electrospinning [37,38,39]. One study also reported a wound dressing material composed of an electrospun PU membrane [40]. Despite the excellent properties of conventional PUs, the PU electrospinning process requires the use of a considerable amount of volatile organic solvent. Although organic solvents play a critical role in the formation and favorable characteristics of electrospun fibers [41], they also cause problems with respect to spinnability and fiber morphology. If a PU-based material is intended to be loaded with certain drugs, then organic solvents pose a risk to the structure of the drugs. In addition, organic solvents have corrosive effects that can cause erosion to the collector of the equipment [42]. The production of solvent-based PUs has been restricted in numerous countries because of toxicity and pollution concerns. For this reason, various studies have proposed WPU formulations [37,40,43]. In the preparation for WPU dispersion and electrospinning, water is the major solvent used and the only material that must be evaporated during the drying process. Therefore, the production of WPU rather than other PUs eliminates the use of toxic chemicals and does not yield polluted air or waste water [44]. However, the dispersibility of WPU is insufficient for the packaging of chains through entanglement to form continuous fibers during electrospinning. To solve this problem, a small amount of a high-molecular-weight water-soluble polymer, such as PEO, can be incorporated to promote chain entanglement and to facilitate electrospinning [45]. PEO can then be easily removed at the end of the process using water, PBS, or alcohol extraction. Additionally, PEO is biocompatible. For these reasons, PEO was used in the present study to stabilize the electrospinning process. We successfully prepared WPU NFMs for use as a biomimetic tendon sheath by employing the electrospinning technique with the use of ecofriendly solvents (i.e., water and PEO). The optimal WPU to PEO volume ratio was discovered to be 1:1 because the NFM obtained with this ration had the highest fiber stability, favorable mechanical properties, and an acceptable in vitro degradation time, thus meeting the requirements for application in clinical settings.

The present study is the first to apply WPU as a barrier for reducing peritendinous adhesion. Using water as a solvent is ecofriendly and enables the use of polar polymers and aqueous dispersions. Water is also less destructive than organic solvents, which are commonly used in the electrospinning of conventional PU, to drugs. These characteristics make WPU an ideal drug carrier. The properties of WPU can be tailored by modifying its composition. These features, when considered alongside its nontoxicity and green properties, indicate the broad potential applications of WPU [46,47]. 

The hydrophilic or hydrophobic nature of a material crucially affects the initial cell attachment [48,49]. Studies have revealed that cells adhere, migrate, and proliferate more favorably on moderately hydrophilic surfaces than on hydrophobic ones [49]. WPU is hydrophobic and, thus, reduced the attachment of fibroblasts in the present in vitro study. In addition, electrophoretic measurements have revealed that mammalian cells have a net negative charge [50]. Electrostatic repulsion may further reduce the degree of adhesion between cells and negatively charged surfaces. Because the WPU surface carries COO^−^ and is, thus, negatively charged, fibroblast attachment to WPU is inhibited. Therefore, the significant reduction in fibroblast attachment to the WPU NFMs in this in vitro study could be explained by the hydrophobic nature and negative charge on the surface of the WPU. The synergistic effect of preventing fibroblast penetration and reducing fibroblast attachment constitutes the antiadhesive effect of the WPU NFM in vivo.

## 4. Materials and Methods

### 4.1. Materials

Isophorone diisocyanate (IPDI, product number 4098719), 2,2-bis(hydroxymethyl)propionic acid (DMPA, product number 106615), 1,4-butanediol (product number 493732), poly(ε-caprolactone) diol (PCL diol; Mn = 2000 Da, product number 189421), triethylamine (TEA, product number 471283), methyl ethylketone (MEK), ethylenediamine (product number 03550), PEO (Mn = 900 kDa, product number 189456), MTT (product number M5655), and acetone were purchased from Sigma-Aldrich Chemical Corporation (St Louis, MO, USA). All chemicals were used as received. Antibiotics, trypsin–ethylenediaminetetraacetic acid, Dulbecco’s modified Eagle’s medium (DMEM), and fetal bovine serum (FBS) were purchased from HyClone.

### 4.2. Synthesis and Physicochemical Analyses of WPU

WPU was produced through a water-borne process. The reaction was conducted under a nitrogen atmosphere. The reaction was verified using the dibutylamine back titration method. A PCL diol was selected as the oligodiol soft segment. PCL was first reacted with IPDI in a vessel purged of nitrogen at 75 °C for 3 h, with 0.03% stannous octoate (Sn(Oct)2) used as the catalyst. After prepolymerization, a small amount of MEK and the ionic chain extender DMPA were incorporated to react for a further 1 h. When the temperature of the vessel had cooled to 45 °C, TEA was added to neutralize the carboxylic group of DMPA. Deionized water was used 30 min later to disperse the neutralized prepolymer, and this addition was accompanied by intense stirring. At the end of the process, ethylenediamine was incorporated for chain extension for another 30 min. The reaction was regularly monitored using FTIR. When the infrared absorbance of NCO groups (a band centered at approximately 2260 cm^−1^) disappeared, the reaction was considered complete. The residual MEK and TEA were removed through vacuum distillation. Finally, WPU was suspended in deionized water in the form of nanoparticles with a solid content of approximately 30 wt% PU [51].

### 4.3. Preparation of Electrospun NFMs

In the preparation of WPU nanoparticle dispersions, 5.66 wt% PEO with respect to the total water content was added to produce a mixture for electrospinning. The mixture was magnetically stirred until the PEO was completely dissolved and the dispersion was homogeneous. The dispersion was placed in a syringe with a 23-gauge blunt-end needle, and the syringe was then mounted on a syringe pump (KD Scientific Inc., Holliston, MA, USA). The syringe pump was set at a constant flow of 0.5 mL/h. The WPU fibers were electrospun by applying a voltage (0–30 kV, CZE1000R; Spellman High Voltage Electronics Corp., Hauppauge, NY, USA) to the needle with the setting of 14 kV, and the current output was limited to a few microamperes. The distance between the ground plate (stainless steel sheet on a screen) and needle tip was 24 cm. Electrospun fibers were collected for 8 h on the screen to produce a randomly braided nonwoven WPU NFM sheet [45,52]. A 10 × 10 cm WPU NFM could be collected. Its thickness would be around 0.2–0.25 mm.

### 4.4. Removal of PEO

PEO was removed from WPU NFMs using PBS extraction. The washing process began with the immersion of WPU NFMs in a PBS solution for 24 h at room temperature; after the PBS treatment, the WPU NFMs were air-dried for 24 h [45].

### 4.5. Characterization of Electrospun NFMs

The morphology of the WPU NFMs before and after the removal of PEO was evaluated using SEM (Hitachi S3000N). For each sample with a specific WPU/PEO composition, at least 100 fibers were randomly selected from 10 images to determine the fiber diameters. The mean and standard deviation of the fiber diameter were calculated. The average size and size distributions of the pores were determined. Measurements were performed using ImageJ software.

### 4.6. Confirmation of WPU NFM

The spectra of the WPU NFMs with different WPU/PEO compositions were evaluated before and after the PBS treatment using attenuated total reflection-FTIR (Spectrum 100, PerkinElmer, Waltham, MA, USA).

### 4.7. Degradation Rate of WPU NFM

The degradation rate of the WPU NFMs was determined according to ISO 10993-13. The in vitro degradation rate was evaluated through a complete immersion of the WPU NFMs in the PBS solution. The ratio of the mass of a WPU NFM to the volume of PBS was 0.1 g:10 mL. The NFMs were soaked in PBS that was maintained at 37 ± 1 °C. The specimens were obtained at the following time points: after 0, 10, 30, 50, 70, 100, and 140 days. The samples were rinsed with water and oven-dried before being weighed (ISO10993-13).

### 4.8. Cytotoxicity and Cell Attachment Test and In Vitro Cell Culture

The cytotoxicity of the WPU NFMs was determined according to the protocol in ISO 10993-5. A piece of WPU NFM with a diameter of 1.4 cm and thickness of 0.2–0.25 mm was extracted and placed in 1 mL of a cell culture medium (DMEM mixed with 10% (*v*/*v*) FBS and 1% (*v*/*v*) antibiotics–antimycotics) at 37 °C for 1 day. The extract was then collected for cell culture. A fresh culture medium was used as the control, an extract of aluminum oxide (Al_2_O_3_) was used as the negative control, and an extract of zinc diethyldithiocarbamate was used as the positive control. HDFs isolated from the fresh tissue culture at the 4th to 6th passage were seeded in a 24-well culture plate for 4 h with 1 × 10^4^ cells per well. The cells were cultured with different extracts for 1 day at 37 °C in a humidified 5% CO_2_ incubator. An MTT assay was then employed to quantify the cell viability. The absorbance was measured at 540 nm using an enzyme-linked immunosorbent assay reader (Synergy HT, BioTek, Winooski, VT, USA). The data were normalized and compared with the absorbance of the control, which was set at 100%. 

In the cell attachment study, TCPS was used as the control, while Seprafilm was used as a comparison. Pieces of WPU NFMs and Seprafilm with a diameter of 1.4 cm were sterilized overnight using ultraviolet light. HDFs isolated from the fresh tissue culture at the 4th to 6th passages were seeded onto the surface of the prewet WPU NFM and Seprafilm pieces as well as TCPS in the 24-well culture plates with 1 × 10^4^ cells per well. The cell-seeded WPU NFM and Seprafilm pieces and TCPS were incubated at 37 °C for 4 h to enable cell adhesion to occur. The cell-seeded NFM pieces, Seprafilm pieces, and TCPS were then transferred to a new 24-well plate, after which they were cultured with 1 mL of a fresh culture medium (DMEM containing 10% (*v*/*v*) FBS and 1% (*v*/*v*) antibiotics–antimycotics) in an incubator set at 37 °C with humidified 5% CO_2_. The number of cells attached to the surface of the WPU NFM pieces, Seprafilm pieces, and TCPS after cell attachment (day 0) and after 1 and 4 days was quantified using an MTT assay. The absorbance was measured at 540 nm by using an enzyme-linked immunosorbent assay reader (BioTek Synergy HT). The absorbance of each group was normalized to that of the TCPS control, which was set at 100%.

### 4.9. Animal Study

Thirty-three 3-month-old New Zealand white rabbits (National Laboratory Animal Breeding and Research Center, Taipei, Taiwan) were used. The animal study procedures were performed in accordance with the guidelines of the Institutional Animal Care and Use Committee of Chang Gung Memorial Hospital (project identification code: 2015121610; with date of approval from 01/January/2017 to 30/June/2018). The rabbits were randomly assigned to the surgical control group, Seprafilm group, or W1P1 NFM group. The repair model of the FDP tendon of the hind foot was selected because the mechanism and anatomy of human finger flexor tendons are similar to those of the FDP tendon. In brief, the palmar skin of zone II of the 2nd and 3rd toes was longitudinally incised, and the pulleys were divided, after which the flexor tendons were released from the synovial sheaths. Both slips of the flexor digitorum superficialis tendon were then excised. The transection of the FDP tendon was subsequently performed using a sharp blade, followed by the repair of the transected FDP tendon by using the Modified Kessler technique with 5–0 prolene. In the W1P1 NFM group, a piece of W1P1 NFM (2 × 1 cm^2^) was wrapped around the repaired FDP tendon, whereas in the Seprafilm group, a piece of Seprafilm (2 × 1 cm^2^) was used instead. For the control group, only the PBS solution was applied to the repaired tendon. Because the first 3 weeks are a crucial period for tendon healing in rabbits, the lower legs and feet of the rabbits were immobilized using casts. 

Three weeks after the procedure, the rabbits were euthanized with their feet transected through the ankle joints. The second and third toes were randomly assigned to either a gross observation or a histological section evaluation to determine the severity of adhesion around the repaired FDP tendon. The histological sections were stained using H&E and Masson trichrome blue. 

The ranges of motion of the DIP and PIP joints were measured using a goniometer. The angle measured from full flexion to full extension between the distal phalanx and middle phalanx was defined as the range of motion of the DIP joint. The angle measured from full flexion to full extension between the middle phalanx and proximal phalanx was defined as the range of motion of the PIP joint.

To quantify the distance of tendon excursion, a constant force of 1 N was applied using a material testing machine to pull the repaired FDP tendon out of the synovial sheath from the tip of the toe. The distance that the FDP tendon could glide out of the digit was defined as the distance of tendon excursion.

The pullout force was measured to determine the extent of peritendinous adhesion. The FDP tendon was pulled using a material testing machine at a constant speed from the tip of the toe. The minimum force required to pull the FDP tendon out of the digit against the adhesion and scarring was defined as the pullout force. 

To determine whether there were any adverse effects related to the W1P1 NFM that could hamper the tendon healing process, the breaking force of the tendon was measured. The repaired FDP tendon was harvested and placed on a material testing machine. The minimum force required to tear the repaired tendon apart was defined as the breaking force of the repaired tendon. 

### 4.10. Statistical Analysis

All data are presented as a mean ± standard error of the mean. The data were analyzed using a one-way analysis of variance to compare the means of different groups. A difference was considered statistically significant if *p* was less than 0.05.

## 5. Conclusions

In the present study, in vitro experiments indicated that the WPU NFM reduced the attachment of fibroblasts without posing major cytotoxicity. In vivo experiments using the rabbit flexor tendon repair model demonstrated that less adhesion occurred (as observed through gross observation and histological section analysis) in the WPU NFM group than in the Seprafilm and control groups. Functional evaluations, namely of the DIP and PIP joint range of motion, the distance of tendon excursion, and the pullout and breaking forces, revealed that the WPU NFMs effectively reduced the amount of peritendinous adhesion and did not hinder the healing of repaired tendons.

## Figures and Tables

**Figure 1 ijms-20-01625-f001:**
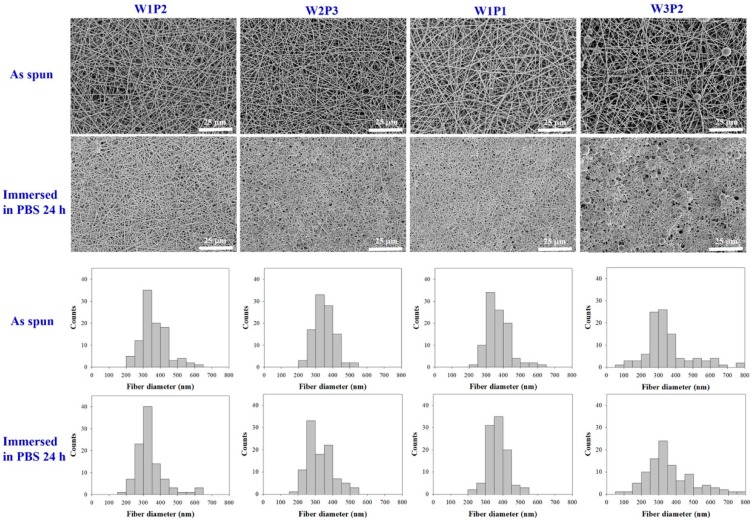
Scanning electron microscopy (SEM) images of electrospun fibers with different compositions of water-borne polyurethane (WPU) and poly-ethylene-oxide (PEO) before and after a phosphate-buffered saline (PBS) treatment: The respective distributions of fiber diameter are plotted as histograms. The SEM images of W3P2 show the bead formation and loss of shape of the fibers after a PBS treatment. W1P1 contains more WPU than W2P3 and W1P2, and its fibers had a preserved shape after the PBS treatment, indicating the stability of the fibers.

**Figure 2 ijms-20-01625-f002:**
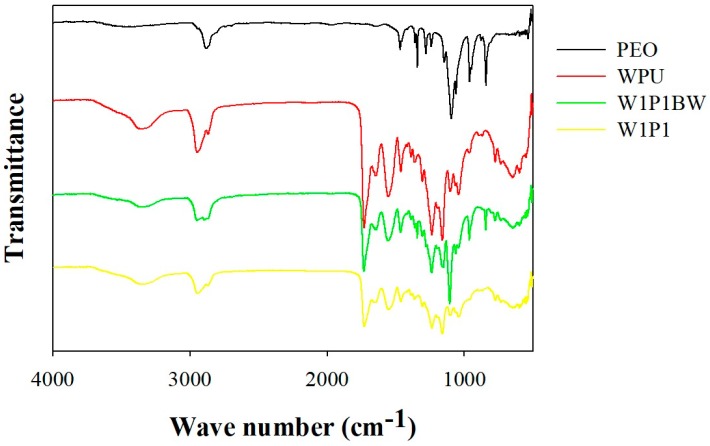
The Fourier transform infrared spectra of the W1P1 nanofibrous membrane (NFM) before and after PBS treatment: A peak centered at approximately 1120 cm^−1^ is identified for W1P1BW (W1P1 before wash) and is attributed to the C–O–C stretching vibration of PEO. This peak was negligible in the spectrum obtained after the NFM was washed with PBS (W1P1). In addition, the spectrum of W1P1 revealed the same absorbance as that in the spectrum of pure WPU, implying that washing with PBS was effective for removing PEO from WPU NFMs.

**Figure 3 ijms-20-01625-f003:**
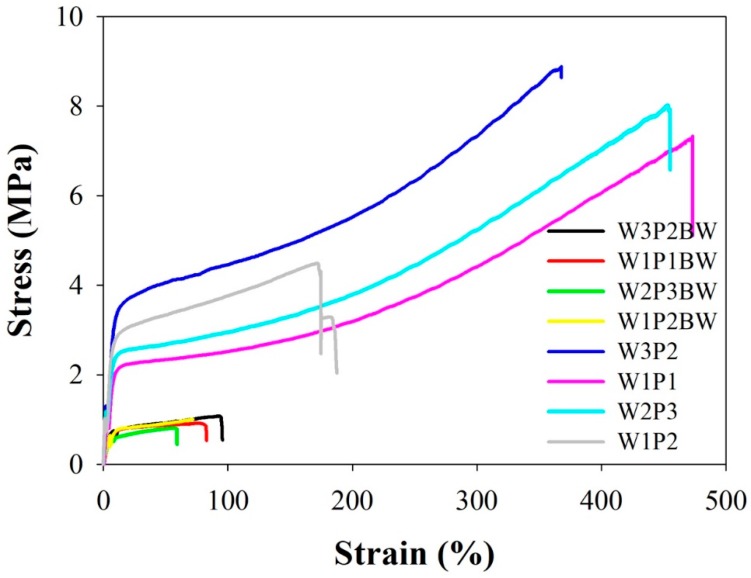
The stress–strain curves of NFMs with different volume WPU/PEO ratios before and after PBS treatment (BW stands for “before wash”; W3P2BW means W3P2 before washing; and W3P2 indicates the status after washing): After PEO was removed through the PBS treatment, the ultimate tensile strength, the elongation percentage at break, and Young’s modulus of all WPU NFMs were higher.

**Figure 4 ijms-20-01625-f004:**
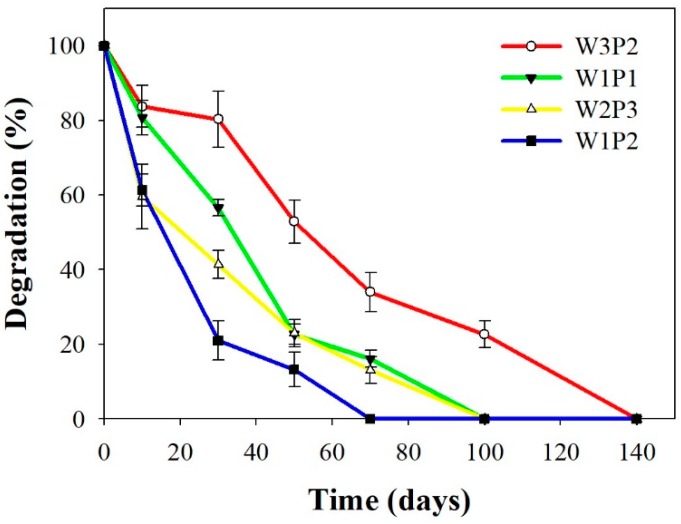
The in vitro degradation curves obtained at 30 °C for NFMs with different WPU/PEO volume ratios: W1P1 degradation took approximately 100 days.

**Figure 5 ijms-20-01625-f005:**
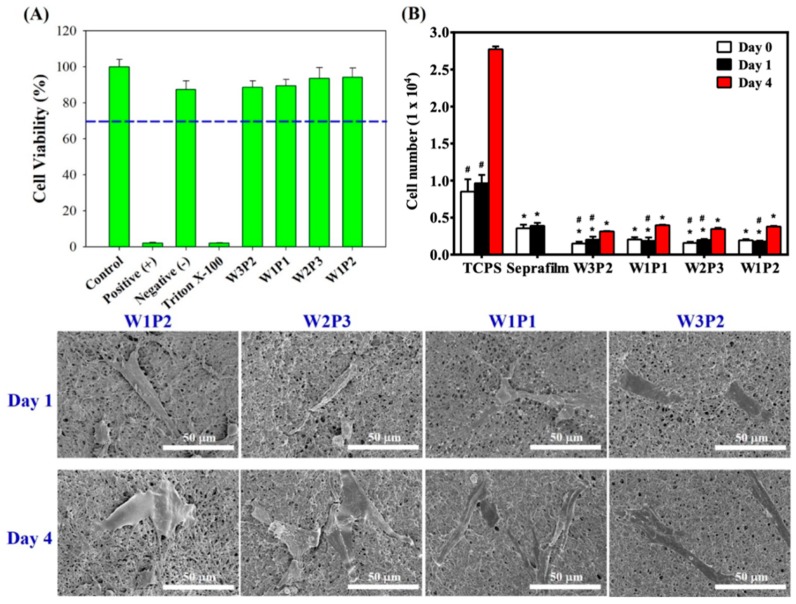
(**A**) The evaluation of cytotoxicity using MTT assays: NFMs with different WPU/PEO volume ratios had similar cell viabilities to the control and negative control, implying that the WPU NFMs are not toxic to human dermal fibroblasts (HDFs). (**B**) The HDF attachment and proliferation assay: HDFs were inoculated on different NFMs and tissue culture polystyrene (TCPS). The cell numbers on day 0, day 1, and day 4 were observed and evaluated using MTT assay. (Bottom rows) The SEM observation of the HDFs attached to NFMs, showing that the HDFs occupied a small area on all NFMs, which correlates with the findings presented in Figure 5B. * *p* < 0.05 when compared with TCPS, # *p* < 0.05 when compared with Seprafilm.

**Figure 6 ijms-20-01625-f006:**
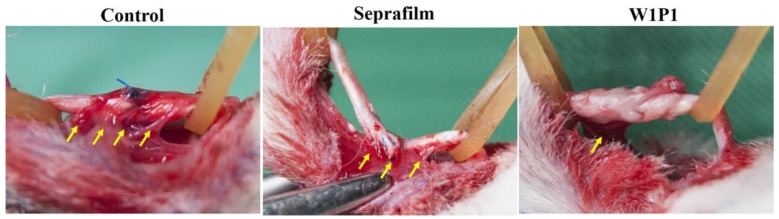
Gross observations of the repaired flexor digitorum profundus tendons in different groups: The yellow arrows indicate the sites of scar adhesion. Severe adhesion and scarring were noted in the surgical control group. For the tendons wrapped with Seprafilm, the adhesion was less severe compared with the surgical controls. The W1P1 NFM group exhibited the least adhesion and scarring around the repaired tendon.

**Figure 7 ijms-20-01625-f007:**
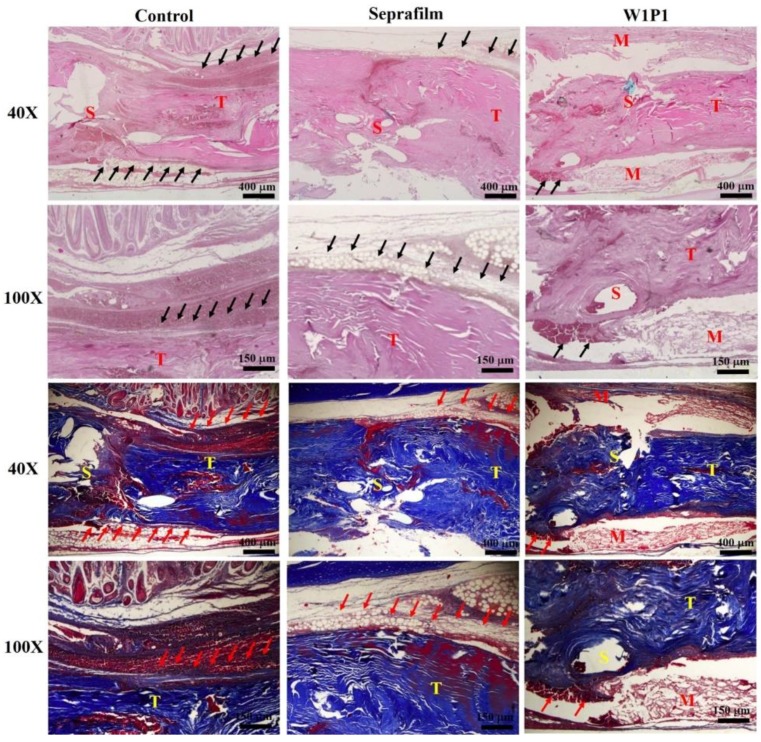
Histological sections stained with hematoxylin and eosin (H&E) (upper two rows) and Masson trichrome blue (lower two rows) of tendons receiving different treatments (S, surgical repair site; T, tendon; and M, material): The black (red) arrows indicate the scar tissue in H&E (Masson trichrome blue) stains. The control group exhibited the most scar tissue around the tendon, whereas the W1P1 group exhibited the least. The NFM can be seen in the W1P1 image, indicating that the material lasted more than 3 weeks. In the Seprafilm group, no remnants of the material were found through histological examination.

**Figure 8 ijms-20-01625-f008:**
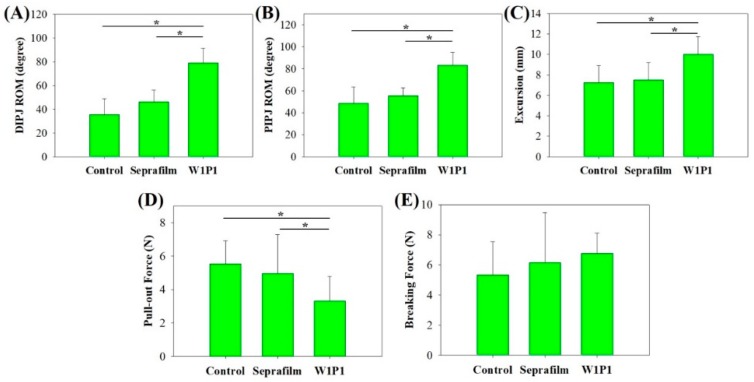
The quantitative evaluations of the antiadhesion efficacy in different groups in vivo: The range of motion (ROM) of the (**A**) distal interphalangeal (DIP) and (**B**) proximal interphalangeal (PIP) joints. The W1P1 group had a significantly greater ROM than the control and Seprafilm groups. (**C**) Tendon excursion, which is the distance that the repaired tendon could glide out of the adhesion, was significantly greater in the W1P1 group than the other two groups. (**D**) The pullout force, which is the force required to completely drag the FDP tendon out of the tendon sheath and its surrounding adhesion, was positively correlated with adhesion severity. Tendons treated with the W1P1 NFM required the least force and significantly less force than the other tendons. (**E**) The breaking force, which is the force required to tear the tendon apart by pulling its ends, was correlated with the degree of healing of the tendon. A greater breaking strength was noted in the W1P1 group than the other groups, but the differences were nonsignificant. * *p* < 0.05.

**Table 1 ijms-20-01625-t001:** The average electrospun fiber diameter and pore size of fibers obtained using different volume ratios (*v*/*v*) of WPU to PEO: The fiber diameters after treatment in PBS for 24 h are also shown. There was no significant difference in the fiber diameter (*p* > 0.05) among different groups before or after PEO removal by PBS immersion. W1P2 and W2P3 showed significantly smaller pore sizes when compared to W3P2.

Groups	Fiber Diameter (nm)	Fiber Diameter after Immersed in PBS 24 h (nm)	Pore Size (µm)
W1P2	362.3 ± 81.4	339.3 ± 83.4	0.78 ± 0.31 *
W2P3	349.5 ± 63.3	324.5 ± 78.1	0.84 + 0.31 *
W1P1	369.3 ± 70.2	368.7 ± 56.3	0.95 ± 0.36
W3P2	352.4 ± 131.5	372.1 ± 159.5	1.05 ± 0.44

* *p* < 0.05 when compared with W3P2.

**Table 2 ijms-20-01625-t002:** The ultimate tensile strength, elongation percentage at break, and Young’s modulus of NFMs with different volume WPU/PEO ratios before and after the PBS treatment (BW stands for “before wash”; W3P2BW means W3P2 before washing; and W3P2 indicates the status after washing). After PEO removal through the PBS treatment, the mechanical properties of all WPU NFMs were improved.

Membrane	Ultimate Tensile Strength (MPa)	Elongation at Break (%)	Young’s Modulus (MPa)
W3P2BW	1.00 ± 0.14	84.8 ± 22.7	0.14 ± 0.03
W1P1BW	0.82 ± 0.09	63.6 ± 16.2	0.13 ± 0.01
W2P3BW	0.82 ± 0.02	41.4 ± 14.8	0.11 ± 0.03
W1P2BW	0.79 ± 0.19	89.8 ± 25.2	0.13 ± 0.04
W3P2	8.00 ± 1.25 ^*,#,‡^	411.69 ± 85.0 ^*,#^	0.68 ± 0.17 ^*^
W1P1	7.25 ± 0.91 ^*,§^	432.4 ± 66.6 ^*,#^	0.64 ± 0.13 ^*^
W2P3	8.58 ± 0.79 ^*#,‡^	445.0 ± 68.8 ^*,#^	0.85 ± 0.30 ^*,‡^
W1P2	4.80 ± 1.58 ^*,§^	236.5 ± 109.1 ^*^	0.92 ± 0.37 ^*,‡^

* *p* < 0.05 compared with the non-PBS-treated groups (W3P2BW, W1P1BW, W2P3BW, and W1P2BW); ^#^
*p* < 0.05 compared with W1P2; ^§^
*p* < 0.05 compared with W2P3; ^‡^
*p* < 0.05 compared with W1P1.

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
