# Peer review of "Electrospun Water-Borne Polyurethane Nanofibrous Membrane as a Barrier for Preventing Postoperative Peritendinous Adhesion"

_ijms, 2019, doi:10.3390/ijms20071625_

Round 1

Reviewer 1 Report

This is an excellent study on the development of a nanofibrous membrane for preventing scar formation and peritendinous adhesion after tendon surgery. 

The manuscript is well-written and well-referenced. The methods are described sufficiently and the results are presented in scientific manner. I have in principle no major issues with this work, just the very minor comments below.

- I suggest to check the references once more. It seems there are some style errors. 

- I recommend to add the product numbers of the chemicals used.

- Please add information into the text why dermal fibroblasts were used? 

Author Response

Reply to Reviewer 1 

1. I suggest to check the references once more. It seems there are some style errors. 
Ans: Sir, we have checked the references and corrected the style errors. Thank you for reminding.

2. I recommend to add the product numbers of the chemicals used.
Ans: Dear Sir, we have added the product numbers of the chemicals used.

3. Please add information into the text why dermal fibroblasts were used? 
Ans: Yes, Sir. Human dermal fibroblasts were used because of two reasons:
1. Peritendinous adhesions are related to extrinsic healing process mediated by extrinsic fibroblasts outside the traumatized tendon.
2. Using human fibroblasts would be more similar to the clinical situation.
These are also added in the manuscript Section 2.4 (first paragraph) and highlighted in yellow.

Reviewer 2 Report

This manuscript presents data supporting the development and characterization of electrospun water-borne polyurethane nanofibrous membranes designed to be used as barriers to prevent post-operative peritendinous adhesions. The manuscript presents physical characterization of the scaffolds as well as both in vitro and in vivo data. Although the manuscript is thorough and presents convincing data, it needs some revisions as outlined below that could strengthened the manuscript before it can be accepted in IJMS:

The      manuscript needs some English editing. There are grammatical errors.

The      abstract could use the description of some of the quantitative data.   Also the first “NFM” abbreviation      should be deleted since it appears after WPU.

The      description of the results is too brief and I encourage the authors to      include more descriptive details that are quantitative in nature. Simply      stating “lower” or “greater” is not good enough.  Be specific in your description, i.e. by      how much was significantly lower or by how much was it greater, etc.

While the      authors state in lines 101/102 that “Using      1/1 PEO/WPU composition (W1P1), there was no bead formation and the fibers      preserved their shape after PBS treatment, indicating their stability”,      the same is true for W1P2 and W2P3. So why choose W1P1?

Figure      1, why show the PBS immersion data? What is the purpose of this especially      when you have degradation data included in the manuscript.

Table      1, no statistical significance is presented and thus describing the      results as lower or higher does not make them significant. The authors      should perform a statistical analysis and show the p value. The same is      true for Table 2, a statistical test is needed to analyze the data.

Line      164/165 – W1P1 is not the only scaffold that perform this way, W2P3 is      almost identical to W1P1. So why choose W1P1? Again, no statistical      analysis is presented and any time point.

Figure      5A and B, no statistical analyses are presented – this needs to be done. Also,      the TCPS is not the ideal control for this experiment. A better control      could have been another type of electrospun membrane that is known for      strong cell adhesion or better yet, Seprafilm since it was used as a      control in the in vivo studies.       Obviously, TCPS is designed for maximal cell adhesion and therefore      to conclude that there is less adhesion on the electrospun scaffolds is      really not a fair comparison. Also, the authors should have included a      zero time point in their proliferation assay.  Typically, to establish the baseline, it      is always customary to include a time zero point to indicate the start of      the experiment where theoretically the cell number are equal between all      different samples. This indicates that everything is the same between all      groups at the beginning of the experiment. Simply showing the data from a      few days later without the initial time point does not mean that      everything was the same at the beginning of the experiment.    

Figures      7, a higher magnification of the area showing adhesions should be included      for all three samples. This is important as it will show the details of      the tissue adhesion.

In the      discussion, the authors mentioned SurgiWrap, why wasn’t this tested in the      in vivo experiments since is thicker and degrades slower than Seprafilm?

Section      4.1, the size and type of font needs to be fixed.

Section      4.3, what were the final dimensions of the NFM? Length, width? Thickness?

Section      4.7. Why was the degradation was done at 300C and not the      customary 370C? A justification is needed.

Section      4.8. What was the thickness of the membrane?

Author Response

Reply to Reviewer 2 

1.    The manuscript needs some English editing. There are grammatical errors.
Ans: Dear Sir, we have sent the article for English editing. Thank you for your suggestions.

2.    The abstract could use the description of some of the quantitative data. Also the first “NFM” abbreviation should be deleted since it appears after WPU.
Ans: Sir, we have corrected the abbreviation in line 23of the abstract. We also added some quantitative data in the abstract and highlighted in green.

3.    The description of the results is too brief and I encourage the authors to include more descriptive details that are quantitative in nature. Simply stating “lower” or “greater” is not good enough.  Be specific in your description, i.e. by how much was significantly lower or by how much was it greater, etc.
Ans: Dear Sir, thank you for the suggestions and we apologize for our mistakes. We have corrected our description. Quantitative data were added in the results and highlighted in green.

4.    While the authors state in lines 101/102 that “Using 1/1 PEO/WPU composition (W1P1), there was no bead formation and the fibers preserved their shape after PBS treatment, indicating their stability”, the same is true for W1P2 and W2P3. So why choose W1P1?
Ans: Sir, during the electrospinning process, W1P1, W1P2, and W2P3 are stable compositions. However, later in the results, we also mentioned that based on the results of mechanical tests, W1P2 is not favorable because of poor tensile strength and elongation percentage at break. 
Therefore, we thought W1P1 and W2P3 might be the better compositions. W2P3 exhibited better ultimate tensile strength than W1P1, though their elongation percentages at break were comparable. In the end, because W1P1 had higher proportion of WPU than W2P3, we choose W1P1 as our optimal composition.
These have been clarified in the results in “2.2 Mechanical properties” as well.

5.    Figure 1, why show the PBS immersion data? What is the purpose of this especially when you have degradation data included in the manuscript.
Ans: Sir, the purpose of PBS immersion was to remove PEO from the membranes (described in line 103~104 and Materials and Methods “4.4 Removal of PEO”). 

6.    Table 1, no statistical significance is presented and thus describing the results as lower or higher does not make them significant. The authors should perform a statistical analysis and show the p value. The same is true for Table 2, a statistical test is needed to analyze the data.
Ans: Sir, thank you for the suggestions. We have performed statistical analysis and added the pvalues in table 1 and 2. The changes have been highlighted in green.

7.    Line 164/165 – W1P1 is not the only scaffold that perform this way, W2P3 is almost identical to W1P1. So why choose W1P1? Again, no statistical analysis is presented at any time point.
Ans: Sir, as the answer for question 4, W2P3 exhibited better ultimate tensile strength than W1P1 (p<0.05), though their elongation percentages at break were comparable  (p>s0.05). In the end, because W1P1 had higher proportion of WPU than W2P3, we choose W1P1 as our optimal composition.
These have been clarified in the results in “2.2 Mechanical properties” as well.
We apologize for not making these clear in the previous version.

8.    Figure 5A and B, no statistical analyses are presented – this needs to be done. Also, the TCPS is not the ideal control for this experiment. A better control could have been another type of electrospun membrane that is known for strong cell adhesion or better yet, Seprafilm since it was used as a control in the in vivo studies. Obviously, TCPS is designed for maximal cell adhesion and therefore to conclude that there is less adhesion on the electrospun scaffolds is really not a fair comparison. Also, the authors should have included a zero time point in their proliferation assay. Typically, to establish the baseline, it is always customary to include a time zero point to indicate the start of the experiment where theoretically the cell number are equal between all different samples. This indicates that everything is the same between all groups at the beginning of the experiment. Simply showing the data from a few days later without the initial time point does not mean that everything was the same at the beginning of the experiment.    
Ans: Dear Sir, thank you for the precious advice. Yes we think that your concern is very important. We revised the figure and added:
1. Day 0 time point;
2. Seprafilm for comparison. 
However, Seprafilm melted around day 3~4, which was difficult to perform the experiment on day 4; therefore, in the figure there is only data for day 0 and day 1 in the Seprafilm group.

9.    Figures 7, a higher magnification of the area showing adhesions should be included for all three samples. This is important as it will show the details of the tissue adhesion.
Ans: Sir, we have updated Figure 7 and added photos of 100X magnification power in addition to the previous 40X ones.

10.  In the discussion, the authors mentioned SurgiWrap, why wasn’t this tested in the in vivo experiments since is thicker and degrades slower than Seprafilm?
Ans:  Based on a reference published in British Journal of Surgery:
Gruber-Blum S, Petter-Puchner AH, Brand J, Fortelny RH, Walder N, Oehlinger W, Koenig F, Redl H (2011) Comparison of three separate antiadhesive barriers for intraperitoneal onlay mesh hernia repair in an experimental model. Br J Surg 98(3):442–9
In this study, a control group and three types of antiadhesive films (SurgiWrap®, Prevadh®, and Seprafilm®) were compared for intraperitoneal hernia repair in a rat model. The surgical control (without any kind of barrier) exhibited severe adhesions in all animals. The use of Prevadh and Seprafilm significantly reduced adhesion formation at 30 days after surgery. However, SurgiWrap® did not have a significant antiadhesive effect, and its degradation was faster than predicted from the manufacturer’s information (6–8 weeks). SurgiWrap® also remained stiff when hydrated and appears inadequate for folding in certain circumstances, such as laparoscopy.

Another article also showed similar result:
ÇİPE, G., KÖKSAL, H. M., of, S. Y. T. J.2011. Efficacy of hyaluronic acid-carboxymethyl cellulose membrane (Seprafilm®) and polylactic acid barrier film (Surgiwrap™) for the prevention of adhesions after thyroid surgery: an experimental model.  Turk J Med Sci 2011; 41(1): 73-9.
In this study, a control group and two types of antiadhesive films (
SurgiWrap®and Seprafilm®) were compared for thyroid surgery in a rat model. The rats were re-operated for observation of adhesion on day 28. They found that Seprafilm® significantly reduced the adhesion. However, in SurgiWrap®group, the adhesion was still severe.

Although the above studies were not dedicated for extremities and tendons, their results imply that Seprafilm® 
might be a superior product compared to SurgiWrap®. Therefore, we used Seprafilm only.

11.  Section 4.1, the size and type of font needs to be fixed.
Ans: Sir, we have corrected the size and type of font. Thank you for reminding.

12.  Section 4.3, what were the final dimensions of the NFM? Length, width? Thickness?
Ans: We usually collect the membrane with a dimension of 10 x 10 cm. The thickness was between 0.2 ~ 0.25 mm. This was also described in the revised manuscript and highlighted in green.

13.  Section 4.7. Why was the degradation was done at 300C and not the customary 370C? A justification is needed.
Ans: We apologize for the typo. It was performed under 370C. We have corrected the manuscript and highlighted it in green.

14.  Section 4.8. What was the thickness of the membrane?
Ans: The thickness of the membrane was 0.2~0.25 mm. This is clarified in the Section 4.8. in the revised manuscript. Thank you.

Round 2

Reviewer 2 Report

Thank you for addressing my comments.